# Cytochrome P450 1B1: A Key Regulator of Ocular Iron Homeostasis and Oxidative Stress

**DOI:** 10.3390/cells11192930

**Published:** 2022-09-20

**Authors:** Yong-Seok Song, Andrew J. Annalora, Craig B. Marcus, Colin R. Jefcoate, Christine M. Sorenson, Nader Sheibani

**Affiliations:** 1Departments of Ophthalmology and Visual Sciences, School of Medicine and Public Health, University of Wisconsin-Madison, Madison, WI 53705, USA; 2Department of Environmental and Molecular Toxicology, Organ State University, Corvallis, OR 97331, USA; 3Department of Cell and Regenerative Biology, School of Medicine and Public Health, University of Wisconsin-Madison, Madison, WI 53705, USA; 4Department of Pediatrics, School of Medicine and Public Health, University of Wisconsin-Madison, Madison, WI 53705, USA; 5Department of Biomedical Engineering, University of Wisconsin-Madison, Madison, WI 53705, USA

**Keywords:** CYP1B1, oxygen induced ischemic retinopathy, congenital glaucoma, retinal endothelial cells, BMP6, hepcidin, iron metabolism, oxidative stress, reactive oxygen species

## Abstract

Cytochrome P450 (CYP) 1B1 belongs to the superfamily of heme-containing monooxygenases. Unlike other CYP enzymes, which are highly expressed in the liver, CYP1B1 is predominantly found in extrahepatic tissues, such as the brain, and ocular tissues including retina and trabecular meshwork. CYP1B1 metabolizes exogenous chemicals such as polycyclic aromatic hydrocarbons. CYP1B1 also metabolizes endogenous bioactive compounds including estradiol and arachidonic acid. These metabolites impact various cellular and physiological processes during development and pathological processes. We previously showed that CYP1B1 deficiency mitigates ischemia-mediated retinal neovascularization and drives the trabecular meshwork dysgenesis through increased levels of oxidative stress. However, the underlying mechanisms responsible for CYP1B1-deficiency-mediated increased oxidative stress remain largely unresolved. Iron is an essential element and utilized as a cofactor in a variety of enzymes. However, excess iron promotes the production of hydroxyl radicals, lipid peroxidation, increased oxidative stress, and cell damage. The retinal endothelium is recognized as a major component of the blood–retinal barrier, which controls ocular iron levels through the modulation of proteins involved in iron regulation present in retinal endothelial cells, as well as other ocular cell types including trabecular meshwork cells. We previously showed increased levels of reactive oxygen species and lipid peroxidation in the absence of CYP1B1, and in the retinal vasculature and trabecular meshwork, which was reversed by administration of antioxidant N-acetylcysteine. Here, we review the important role CYP1B1 expression and activity play in maintaining retinal redox homeostasis through the modulation of iron levels by retinal endothelial cells. The relationship between CYP1B1 expression and activity and iron levels has not been previously delineated. We review the potential significance of CYP1B1 expression, estrogen metabolism, and hepcidin–ferroportin regulatory axis in the local regulation of ocular iron levels.

## 1. Introduction

Cytochrome P450s (CYPs), named after the absorbance peak near 450 nm of their Fe(II)-carbon monoxide complex in rat liver microsomes [1], are a superfamily of heme-containing enzymes. The CYP superfamily includes approximately 9000 proteins classified into more than 800 families, which makes CYPs one of the largest and most functionally diverse protein superfamilies [2]. In humans, there are 57 putatively functional genes and 58 pseudogenes arranged into 18 CYP families and 43 subfamilies, distributed over most autosomal chromosomes [3]. In comparison, there are 108 functional and 88 pseudogenes representing CYPs in mice [4]. There are also many known alternatively spliced CYPs that further expand the number of functional CYP proteins [5] (Table 1). CYPs are membrane-bound enzymes, mostly found in the smooth endoplasmic reticulum and some in mitochondria. The enzymes are best known to catalyze a monooxygenation reaction (RH + O_2_ + NAD(P)H + H^+^ → ROH + H_2_O + NAD(P)^+^, where RH stands for a substrate with a hydroxylatable site [6] (Figure 1). CYPs are often referred to as poly-substrate monooxygenases. 

CYPs also catalyze a variety of other reactions, such as reduction, desaturation, ester cleavage, and rearrangement of fatty acids [9]. CYP substrates include exogenous chemicals such as drugs, food toxicants, and carcinogens, as well as endogenous compounds, for example, steroids, prostaglandins, and bile acids. As the main site of metabolism of exogenous chemicals in humans, the liver expresses around 30 CYPs out of the 57 identified in humans [10]. Among CYP enzymes in the human liver, CYP3A4 is the most abundant and represents approximately 22.1% of the total CYP enzyme protein content [11]. Along with CYP3A4, CYP1A2, CYP2C9, CYP2C19, CYP2D6, and CYP3A5 are responsible for the biotransformation of about 80 percent of all marketed drugs [12]. CYP enzymes are also expressed throughout the body, and emerging evidence suggests that extrahepatic CYP enzymes, such as CYP1B1, have important roles in modulating tissue metabolic homeostasis, developmental processes, and contributing to carcinogenesis and other environmental diseases.

### CYP1B1

CYP1B1, one of the CYP enzymes, is mainly expressed in extrahepatic tissues [4] and is the only member of the CYP1B subfamily. It was initially identified from mouse embryonic fibroblasts (C3H10T1/2) following incubation with benzo(a)anthracene and 2,3,7,8-tetrachlorodibenzo-p-dioxin (TCDD) [13,14]. The human CYP1B1 was cloned from primary human skin keratinocytes shortly after [15]. Although CYP1B1 is assigned to the CYP1 family, CYP1B1 shows a low degree of genetic homology (~40%) with other CYP1 family enzymes, CYP1A1 and CYP1A2 [16], and is a significantly larger protein than other human CYPs. CYP1B1 is expressed in various adult tissues such as bone marrow, brain, breast, intestine, kidney, prostate, and ocular tissues. CYP1B1 expression is also found during development of the hindbrain, neural crest, and eyes in early human and mouse embryos [17,18] and has an important role in the generation and action of retinoic acid [19,20]. Early developing hepatocytes also express CYP1B1, which significantly diminishes with liver maturation [21]. Although CYP1B1 expression is constitute in most tissues and cells, its expression is also regulated by various hormones, cell–cell adhesion, and aryl hydrocarbon receptor (AhR) ligands [22,23,24,25]. CYP1B1 expression affects liver stellate cell activation and cholesterol synthesis in neonatal liver [20]. The 3’-untranslated region of CYP1B1 is also a target of various miRNAs whose regulation could impact CYP1B1 expression in various tumor cells [26].

CYP1B1 metabolizes a range of compounds, including exogenous chemicals such as polycyclic aromatic hydrocarbons, dioxins, and aflatoxin B1. CYP1B1 also metabolizes endogenous bioactive compounds, including estradiol, arachidonic acid, vitamin A, and melatonin (reviewed in [21,27]). These metabolites are mediators of various cellular and physiological processes [28,29]. Thus, CYP1B1 expression and function could be important in the proper development and maintenance of metabolic homeostasis in various organs and tissues. In fact, mutations of the human CYP1B1 gene are the most common causes of autosomal recessive primary congenital glaucoma in the world [30]. However, the underlying molecular and cellular mechanisms involved remain largely unknown.

CYP1B1 expression is found in various ocular tissues, including ciliary body, iris, cornea, trabecular meshwork, and retina [31,32,33]. Using *Cyp1b1*-deficient (*Cyp1b1*^−/−^) mice, our group reported constitutive expression of CYP1B1 in retinal vascular cells, including astrocytes, pericytes, and endothelial cells, as well as trabecular meshwork cells of the conventional outflow pathway. These studies established a significant role for CYP1B1 expression and activity in the regulation of redox homeostasis in retinal vasculature [34,35,36] and trabecular meshwork [32,37]. We showed, in the absence of CYP1B1, there is a significant increase in the oxidative stress in the retina as well as in retinal and trabecular meshwork cells in vitro. In addition, none of the commonly recognized sources of oxidative stress, including NADPH-oxidase, lipoxygenases, mitochondria, and xanthine oxidase, appeared to be involved [34]. We also showed that N-acetylcysteine was the most effective antioxidant, among several tested, to restore the normal redox state in vivo and in vitro (Figure 2). Thus, the underlying mechanism(s) as to how CYP1B1 deficiency contributes to altered redox homeostasis in various ocular cells and/or tissues deserves further elucidation. 

In recent studies, we reported that livers from *Cyp1b1*^−/−^ mice exhibit altered expression of 560 genes, including suppression of peroxisome proliferator-activated receptor ɣ (PPARɣ) and many genes regulated by PPARα [38]. The nuclear receptors, PPARs, regulate the expression of several genes and are critical in the lipids and glucose homeostasis. The liver from *Cyp1b1*^−/−^ mice also showed decreased expression of hepcidin, a peptide hormone with important roles in the regulation of systemic iron homeostasis [39,40,41]. Thus, these results suggest a potential role for CYP1B1 in the modulation of systemic iron homeostasis [42,43]. 

We propose that CYP1B1 expression in the retinal endothelium may contribute to the processes involved in local iron homeostasis. It remains unclear how CYP1B1 and CYP1B1 metabolites regulate ocular iron homeostasis. Alterations in the bone morphogenic protein 6 (BMP6)-hepcidin axis in *Cyp1b1*^−/−^ liver, and locally in retinal endothelial cells (EC), could play significant roles in regulating systemic and ocular iron homeostasis altering redox status in the eye. Here, we review the regulation of iron homeostasis as an important mechanism of maintaining ocular redox status by CYP1B1, especially in the retina. We also discuss the potential roles of CYP1B1 and its metabolites in the local regulation of iron levels through the hepcidin–ferroportin axis affecting redox homeostasis in the retina. 

## 2. Iron and Its Biological Functions

Transition metals, whose atoms have partially filled d subshells, play important roles in various biological processes in forms of life from bacteria to humans. Transition metals form ions that exist in multiple oxidation states, which allows them to accept or donate electrons to other molecules. Thus, transition metals including iron, copper, manganese, nickel, and zinc are used as cofactors in many enzymes for normal cellular function. Iron is an essential element for various cellular processes, including cellular respiration, oxygen transport, DNA synthesis, and DNA repair. The importance of iron is due to its efficient ability to gain and lose electrons, which allows iron to be involved in a number of biochemical reactions [44]. Iron is the most abundant metal in the human body [45], and proteins can contain iron as part of cofactors such as protoporphyrin IX iron complex (heme), iron–sulfur (Fe-S) clusters, and iron ions. Iron functions in a redox mode that involves changes between +3 and +2 states. Iron also forms complexes with molecular oxygen that functions as a carrier (hemoglobin) or is converted to FeO (+4 state) reactive intermediates in hydroxylases, often with heme (CYP450s) but sometimes as direct complexes (Figure 1). Fe-S complexes are typically electron transfer proteins. The binding partner determines the redox potential as a consequence of favoring +3 or +2 state.

Heme is an iron-containing porphyrin and constitutes 95% of functionally utilized iron in the human body [45]. In various types of cells, heme is involved in oxygen transport, oxygen storage, and electron transfer. Heme-containing proteins include hemoglobin, cytochrome bc1 complex, cytochrome c oxidase, succinate dehydrogenase, prostaglandin endoperoxide synthase, nitric oxide synthase, and cytochrome P450s [46,47]. Heme also acts as a signaling molecule that regulates multiple molecular and cellular processes, including signal transduction and protein complex assembly. For example, the transcription repressor BTB and CNC homology 1 (Bach1) is a mammalian transcription factor that has heme-binding sites. The binding of heme to Bach1 inhibits its DNA-binding activity and promotes Bach1 nuclear export [48].

Iron–sulfur clusters, considered as one of the earliest types of protein prosthetic groups, are polynuclear combinations of iron and sulfur atoms. In proteins, they mediate a multitude of functions in different cellular organelles, and their key functions are electron transfer and redox reactions. Iron–sulfur cluster-containing proteins are important players in mammalian metabolism, including oxidative phosphorylation, the citric acid cycle, and nucleic acid metabolism. Electron-transfer flavoproteins, NADH: ubiquinone oxidoreductase, Rieske iron–sulfur protein, and succinate dehydrogenases are examples of iron–sulfur cluster-containing proteins, and they are involved in mitochondrial respiratory chain function [47]. Aconitase, an enzyme that catalyzes the isomerization of citrate to isocitrate in the citric acid cycle, contains a [3Fe-4S]^1+^ cluster that can be converted to an active [4Fe-4S]^2+^ at its active site [49]. In nucleic acid metabolism, a number of enzymes including the DNA helicase XPD and its paralogues, such as FancJ (Fanconi’s anemia complementation group J), RTEL1 (regulator of telomere length 1), and all nuclear replicative DNA polymerases, contain iron–sulfur clusters [50].

In addition to iron-containing prosthetic groups, iron-ion is found as a cofactor in proteins. These enzymes include diiron monooxygenases, such as δ-9-fatty acid desaturase and the beta (small) subunit of ribonucleotide reductase [51]. Moreover, hypoxia-inducible factor (HIF) proline dioxygenases, also known as prolyl 4-hydroxylases, require Fe(II) and 2-oxoglutarate (also known as α-ketoglutarate) and ascorbate to hydroxylate HIF1α and trigger its degradation [52]. In mitochondria, the Fe(II) and 2-oxoglutarate-dependent oxygenase Alkb homolog (ALKBH1) is involved in the regulation of mitochondrial-tRNA modification [53], and ALKBH7 mediates cellular necrosis through the modulation of glyoxal metabolism [54].

## 3. Iron Homeostasis

Iron absorption, transport, distribution, and storage are tightly regulated by several specific proteins and are discussed below. On average, a 70 kg man contains 3.5–4 g of iron. Most of the iron is intracellular and carried in molecules, such as the hemoglobin of red blood cells (about 2.0–2.5 g), ferritin in hepatocytes and macrophages (about 0.5–1 g), and in myoglobin, ferritin, and iron-containing enzymes in other types of cells (about 0.5 g in total). Only a few mg of iron is contained in blood plasma, and the majority of it is bound to transferrin [55]. Unlike most other essential nutrients, there are no regulated mechanisms for the excretion of iron in mammals. Iron excretion results from the exfoliation of dead skin, blood loss, and turnover of intestinal epithelial cells, which are independent of iron levels in the body. In humans, about 1–2 mg of iron is lost every day, and the loss is balanced by iron absorption, which occurs in the duodenum and proximal jejunum. Due to the lack of controlled iron excretion, systemic iron homeostasis is achieved mainly through the regulation of absorption, utilization, and recycling of iron [56].

Enterocytes of the proximal small intestine uptake dietary iron via divalent metal-iron transporter-1 (DMT1) (Figure 3A). Located on the apical membrane of enterocytes, DMT1 only transfers Fe^2+^, but most nonheme dietary iron exists as Fe^3+^. Thus, iron uptake through DMT1 requires a reduction in Fe^3+^, mediated by duodenal cytochrome B (DCYTB), an iron-regulated ferrireductase that is highly expressed in the apical membrane of duodenal enterocytes [57]. Dietary heme is imported by heme carrier protein (HCP1) into enterocytes and degraded by heme oxygenase 1 (HO1) to release Fe^2+^, which likely joins the same intracellular iron pool with nonheme iron [58]. Iron transfer from enterocytes into the circulation is mediated by ferroportin located at the basolateral membrane of enterocytes. Ferroportin, also known as iron-regulated transporter 1 or solute carrier family 40 number 1 (SLC40A1), is a transmembrane protein with 12 transmembrane alpha helices. Ferroportin is the only known cellular iron efflux transporter in vertebrates, and it is typically expressed in iron-exporting cells, including enterocytes, macrophages, and hepatocytes [59].

Due to its half-filled 3d^5^ electron configuration, Fe^3+^ is more stable and less water-soluble than Fe^2+^, which is water-soluble and reactive. Exported as Fe^2+^ through ferroportin, iron undergoes oxidation to Fe^3+^. Ferroportin cooperates with a membrane-bound ferroxidase hephaestin, which is located in the basolateral membrane of enterocytes and functions to convert Fe^2+^ to Fe^3+^ [60]. Hephaestin is a membrane-bound homolog of ceruloplasmin, which is an enzyme containing six atoms of copper and is produced in the liver. Secreted from the liver into the systemic circulation, ceruloplasmin functions to oxidize Fe^2+^ in the circulation into Fe^3+^ [61]. Almost all the iron in the circulation binds to transferrin, a 76–80 kDa bilobal glycoprotein that is produced predominantly by the liver. Transferrin contains two binding sites for Fe^3+^ and transports iron through the blood to various tissues. Transferrin-bound iron only accounts for 0.1% (3–4 mg) of the total iron in the body (3–4 g in an adult man) [62]. However, due to the rapid turnover rate of the transferrin-bound iron complex (about 10 times/day), transferrin forms an important iron pool (20–25 mg/day) to meet the daily demands of iron for physiological processes, including erythropoiesis [63,64]. Transferrin delivers iron through the circulation to cells expressing the transferrin receptors, such as retinal vascular EC [65].

**Figure 3 cells-11-02930-f003:**
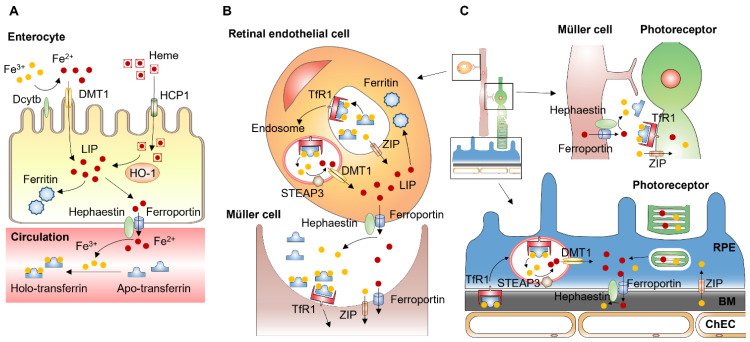
Systemic and local iron uptake and transport. (**A**) Enterocytes uptake dietary iron via divalent metal transporter-1 (DMT1) on the apical membrane. Iron uptake through DMT1 is mediated by duodenal cytochrome B (DCYTB), an enzyme that reduces Fe^3+^ to Fe^2+^. Dietary heme is imported by heme carrier protein (HCP1) into enterocytes and degraded by heme oxygenase 1 (HO1) to release Fe^2+^. Iron transfer from enterocytes into the circulation is mediated by ferroportin located at the basolateral membrane of enterocytes. Ferroportin cooperates with ferroxidase hephaestin converting Fe^2+^ to Fe^3+,^ which binds to transferrin in the circulation. (**B**) Retinal endothelial cells (EC) express transferrin receptor 1 (TfR1) at the apical membrane. After binding of iron-loaded transferrin, the TfR undergoes clathrin-mediated endocytosis. Within the endosome, Fe^3+^ is released from transferrin and reduced to Fe^2+^ by six-transmembrane epithelial antigen of prostate 3 (STEAP3). Fe^2+^ is then transported from the endosome to cytosol by DMT1. Unbound Fe^3+^ can be transported into retinal EC via Zinc transporters (ZIP) such as ZIP8 and ZIP14. Exported from retinal EC, iron is imported by glial cells such as Müller cells. (**C**) Exported by Müller cells, iron can be imported by photoreceptors expressing transferrin receptor, Zip8 and Zip14. Photoreceptors export iron via phagocytosis of shed photoreceptor outer segments by retinal pigment epithelium (RPE) cells. RPE cells also import iron from the choroid via transferrin receptor [66,67]. BM: Bruch’s membrane.

Transferrin receptors are homodimeric transmembrane glycoproteins that mediate the uptake of transferrin-bound iron into the cells. Transferrin receptor 1 (TFR1, also known as CD71) is ubiquitously expressed in mammalian tissues and cells. Expression of TFR1 is regulated by intracellular iron levels. Under conditions of iron deficiency, iron-regulatory proteins (IRP) bind to the iron-responsive element (IRE) motifs in the 3′-untranslated region of TFR1 mRNA to prevent endonucleolytic cleavage, mediating post-transcriptional stabilization of TFR1 mRNA [68]. Transferrin receptor 2 (TFR2) is expressed primarily in the liver and erythroid precursors. TFR2 expression is not regulated by IRPs, as TFR2 does not have IRE motifs in its 5′ and 3′ untranslated regions, which indicates TFR2 expression is not regulated by intracellular iron status [69]. The binding affinity to iron-bound transferrin of TFR2 is 25-fold lower than that of TFR1, which suggests that TFR1 is the major receptor for cellular uptake of iron-bound transferrin. However, TFR2 plays an important role in iron homeostasis through the regulation of BMP signaling pathways. In hepatocytes, TFR2 is one of the auxiliary factors for BMP receptors modulating hepcidin expression, and TFR2 mutations result in reduced hepcidin production [70].

After binding of iron-loaded transferrin, the transferrin receptor undergoes clathrin-mediated endocytosis (Figure 3B). The acidic endosomal pH (~5.6), along with other factors such as conformational changes in transferrin, salt concentration, temperature, and chelators within the endosome, contribute to the release of Fe^3+^ from transferrin [71]. Fe^3+^ in the endosome is reduced to Fe^2+^ by an endosomal membrane ferric reductase, the six-transmembrane epithelial antigen of prostate 3 (STEAP3). The Fe^2+^ is then transported from the endosome to cytosol by the transmembrane protein DMT1 [72]. The redox-active Fe^2+^ is utilized by a variety of cellular enzymes as mentioned above. 

## 4. Liver and Systemic Iron Homeostasis

Liver is one of the most functionally diverse organs in the body. The most common cells in the liver are hepatocytes, which are responsible for 60–70% of the liver cell population or 90% of total liver mass [73]. As the liver parenchymal cells, hepatocytes play significant roles in multiple physiological processes, including detoxification of xenobiotic chemicals, gluconeogenesis, lipid metabolism, digestion, and iron homeostasis. Although mature hepatocytes express very low or no CYP1B1, the other cellular components of liver express significant amounts of CYP1B1, including liver sinusoidal endothelial cells [43]. Liver is the central organ of systemic iron homeostasis. First, liver stores iron in proteins such as ferritin and hemosiderin in hepatocytes [74]. Ferritin is a 450 kDa storage protein and a complex of two subunits, termed H (heavy chain) and L (light chain), which stores up to 4500 atoms of Fe^3+^. Hemosiderin is found in lysosomes, and it is a complex of partially degraded ferritin aggregates, denatured proteins, and lipids. Ferritin accounts for most of iron storage in the normal liver and, in the case of iron overload, more iron is stored in hemosiderin [75]. 

Liver also contributes to systemic iron homeostasis, as hepatocytes produce a 25-amino acid peptide hormone, hepcidin. Released from the liver, hepcidin circulates and binds to the cellular iron exporter, ferroportin [76]. Ferroportin is expressed on iron-exporting cells, including duodenal enterocytes, splenic macrophages, liver Kupffer cells, periportal hepatocytes, and the placental syncytiotrophoblast [77]. Recent studies also suggest that ferroportin and hepcidin are widely expressed and are key modulators of cell autonomous intracellular iron in many ocular cell types, including retinal EC and trabecular meshwork cells [78]. The binding of hepcidin to ferroportin leads to the endocytosis and lysosomal degradation of ferroportin, which downregulates cellular iron efflux. Thus, the ferroportin–hepcidin axis plays a key role in maintaining intracellular and systemic iron homeostasis, and alterations in this regulatory pathway are linked to dysregulation of iron homeostasis and tissue damage. 

Hepatic hepcidin expression is controlled through transcriptional regulation by several stimuli, including iron stores in the liver and the rate of erythropoiesis, inflammation, hypoxia, and oxidative stress. Iron-mediated hepcidin production is regulated via the BMP-suppressors of mothers against decapentaplegic (SMAD) signaling pathways [79]. BMPs are multifunctional cytokines, which belong to the transforming growth factor β (TGFβ) superfamily. Among more than 12 BMPs identified in vertebrates, the hepatic expression of BMP2, 4, 5, 6, and 9, have been reported [80,81]. BMP2, 4, and 9 induce hepcidin expression, but BMP6 appears to be the predominant mediator of hepatic hepcidin expression in response to iron levels in vivo [82]. Iron overload in mice implemented by iron-enrichment diet or inactivation of the *Hfe* gene induces *Bmp6* expression in the liver but not in the duodenum [83]. BMP6-null mice show downregulated hepatic hepcidin expression and increased iron levels in the circulation and tissues, including the liver, heart, and pancreas [84]. BMP6 also regulates retinal iron homeostasis, and its altered expression may contribute to iron accumulation in age-related macular degeneration [85]. 

In the liver, BMP6 is produced by non-parenchymal cells, mainly sinusoidal endothelial cells (SEC). Secreted from SEC, BMP6 acts in a paracrine manner by binding to BMP receptors on hepatocytes. BMP receptors are serine/threonine kinases, which belong to the TGF-β receptor superfamily. Three type I receptors (BMPRIA, BMPRIB, and activin receptor (ACVR) type IA) and three type II receptors (BMPRII, ActRIIA, and ActRIIB) have been identified. In the absence of ligand, type II and type I receptors are located on the cell surface. Upon ligand binding, homomeric dimers of type I and type II receptor form a tetrameric complex [86]. The type II receptors contain a constitutively active cytoplasmic kinase domain, which phosphorylates and activates type I receptors. The activated type I receptors subsequently phosphorylate receptor-regulated SMADs (also termed R-SMADs). R-SMADs include SMAD1, SMAD5, and SMAD8, which are specifically involved in BMP signaling. SMAD2 and SMAD3 are restricted to TGFβ signaling [87]. Phosphorylated R-SMADs form homomeric complexes and bind to the common mediator SMAD4 to form a trimeric complex which acts as a transcription factor. The SMAD complexes translocate into the nucleus, where they participate in transcriptional regulation of target genes through binding to DNA directly or indirectly in conjunction with other coregulator proteins [88]. BMP6 is also expressed by retinal pigment epithelial (RPE) cells, and its downregulated expression in age-related macular degeneration (AMD) likely contributes to iron build up in AMD. This is likely mediated through the increased oxidative stress associated with pathophysiology of AMD. Oxidative stress decreases BMP6 levels, while iron increases BMP6 levels in RPE cells in culture [85]. 

The BMP signaling pathways are involved in a number of cellular mechanisms, and the signaling can be further modulated by co-receptors such as endoglin and repulsive guidance molecule (RGM) protein family, including RGMa, RGMb (also known as DRAGON), and RGMc (also known as hemojuvelin (HJV) or hemochromatosis type 2 protein (HFE2)) [89,90]. BMP co-receptors lack enzymatic activity, but by directly interacting with ligands and type I and type II receptors, they can regulate ligand accessibility to receptors and/or signaling specificity [91]. Hemojuvelin (HJV) is a glycophosphatidylinositol-linked membrane protein and a BMP co-receptor. Hemojuvelin expression is detected in the liver, heart, and skeletal muscle. In the liver, HJV functions as a co-receptor for the BMP receptor complex and triggers the binding of BMP6 to its receptors to induce hepcidin expression. In humans, mutations in the HJV genes are the most common cause of juvenile hemochromatosis, a severe and early-onset form of hereditary iron overload characterized by a profound hepcidin insufficiency [92].

We previously reported decreased expression of hepcidin in livers of *Cyp1b1*^−/−^ mice. Liver is the central organ responsible for systemic iron homeostasis, and liver SEC is responsible for the production and secretion of BMP6. BMP6 acts on neighboring hepatocytes to enhance hepcidin production [93]. However, whether the noted changes in hepcidin levels in livers from *Cyp1b1*^−/−^ mice are mediated by changes in BMP6 expression in the liver SEC was not previously addressed. We recently reported isolation and characterization of liver SEC from *Cyp1b1*^+/+^ and *Cyp1b1*^−/−^ mice [43]. Figure 4 shows a decreased BMP6 and hepcidin expression in liver SEC prepared from *Cyp1b1*^−/−^ mice compared with *CYP1b1*^+/+^ mice. In addition, BMP6 receptor expression, namely Bmpr1a and Bmpr2, were also downregulated in *Cyp1b1*^−/−^ liver SEC. The results presented here are consistent with the reduced expression of hepcidin noted in the liver from *Cyp1b1*^−/−^ mice [42] and likely associated with increased systemic iron levels due to stabilization and increased levels of iron exporter ferroportin. Ferroportin is the target of hepcidin-mediated degradation and the mitigation of systemic release of iron from cells such as enterocytes and macrophages [40]. 

The retina is normally protected from changes in the systemic circulation through the retinal–blood barrier established by the retinal vascular EC (Figure 3B). The retinal vascular EC express molecular mediators of iron regulation, such as hepcidin and ferroportin, and are key determinants of local iron homeostasis in the retina [94]. The increase in systemic iron is shown to decrease ferroportin levels in retinal EC [95], perhaps through upregulation of BMP6 expression. The intraocular administration of BMP6 protein increases retinal hepcidin and alters retinal labile iron levels [85]. Iron is an essential element for various cellular functions due to its ability to transfer electrons, but it catalyzes the formation of hydroxyl radicals from hydrogen peroxide via the Fenton reaction (Fe^2+^ + H_2_O_2_ → Fe^3+^ + · OH + OH^−^). In excess, iron can lead to accumulation of lipid peroxidation generating reactive oxygen species (ROS) and oxidative stress. Thus, ocular iron levels are tightly regulated to maximize iron supply and avoid ROS generation. Alterations in iron homeostasis, such as iron accumulation and iron deficiency, which impact oxidative stress, can lead to cell and tissue damage and dysfunction.

Retinal EC express ferroportin is the iron exporter on the abluminal (basolateral) membrane facing the neuroretina responsible for importing systemic iron to the retina [78]. Ferroportin is also expressed in the basal membrane of RPE and muller cells’ endfeet, which likely mediate the export of iron from the retina to choroid through RPE and vitreous through the muller cells’ endfeet. The deletion of ferroportin in retinal EC resulted in increased iron levels in retinal EC [96], as also demonstrated by intravitreal delivery of hepcidin. Thus, increased expression of hepcidin, as a result of increased BMP6 levels within the retina, could reduce ferroportin expression on retinal EC, resulting in the intracellular accumulation of iron and protection of neuroretina from excess systemic iron levels. 

## 5. Local Iron Homeostasis in the Retina

In the retina, like other tissues, iron is important for cellular metabolisms and is particularly critical for the visual phototransduction cascade. Fe^2+^ is a cofactor for the retinal pigment epithelium 65 kDa protein (RPE65), which is an isomerohydrolase that converts all-trans retinal ester to 11-cis retinal, the light-absorbing chromophore. Photoreceptor cells shed and regenerate disc membranes by using iron-containing enzymes including fatty acid desaturase [97,98]. Although iron is important for catalyzing enzymatic reactions, excessive iron levels result in ROS accumulation leading to oxidative stress in cells and tissues. The retina is one of the most metabolically active tissues, and it is very rich in polyunsaturated fatty acids containing membranes, which are susceptible to lipid peroxidation initiated by free radicals [99]. The accumulation of iron is observed in several retinal diseases. Hereditary and acquired iron overload disorders, including aceruloplasminemia, hereditary hemochromatosis, Friedreich’s ataxia, and secondary hemochromatosis, are associated with retinal degeneration (reviewed in [66]). Thus, the maintenance of optimal levels of iron is required to provide adequate iron for cellular and tissue functions and to avoid oxidative stress caused by excess levels of iron. Studies of rodent models indicate that the control of iron homeostasis in the retina is maintained by locally produced proteins by retinal cells, with Picard and colleagues proposing a hypothetical model of retinal iron uptake and homeostasis (reviewed in [94]).

Proton-induced X-ray emission showed that iron is widely and unevenly distributed throughout the adult (55 days old) rat retina. The highest concentrations of iron were observed in the RPE/choroid and the inner segments of photoreceptors [100]. Iron concentrations were relatively lower, but significant amounts of iron were found in other retinal layers, including the outer segments of photoreceptors, the inner nuclear layer, and the ganglion cell layer. The outer nuclear layer showed the lowest iron concentration in rat retinal layers [101]. Unlike other organs, the retina is protected from the systemic circulation by the blood–retinal barrier (BRB), which is composed of two tight barriers. The inner BRB is primarily formed by the tight junctions of retinal EC lining the microvasculature of the inner retina and contributions by other cell types including pericytes and macroglia (astrocytes and Müller cells). The outer BRB refers to the tight junctions of the RPE cells that comprise a single layer of cells located between the photoreceptors and Bruch’s membrane. Unlike choroidal EC, RPE and retinal EC lack fenestrations, and thus, the BRB prevents passive diffusions of large molecules. Non-heme-bound iron can be transported to the cells via two possible mechanisms: transferrin-bound iron uptake and non-transferrin-bound iron uptake. As most plasma iron is bound to transferrin [55], transferrin-bound iron import is the main transport mechanism for iron entry across the BRB [94]. 

In the outer BRB, RPE cells have been reported to express transferrin receptors. Primary human RPE cells [102] and ARPE-19 [103] cells showed transferrin receptor expression, detected in both basolateral and apical surfaces. This indicates that RPE cells can import iron from photoreceptors (apical side) and choroidal vasculature (basolateral side). However, the localization of transferrin receptors in the RPE cells remains unclear, as in vivo studies showed that in adult mice [104] and neonatal rats [105], transferrin receptors localization is polarized to the basolateral side of the RPE cells. 

Imported iron is exported via ferroportin. In the mouse retina, ferroportin is predominantly detected in the basolateral side of the RPE cells [67], suggesting that iron could be exported from the RPE cells to the choroid as discussed above. In the inner BRB, studies in mice suggest that retinal EC express transferrin receptors on the luminal side of retinal EC [96]. The immunostaining of mouse retina showed that ferroportin localizes at the abluminal side of retinal EC [106]. Based on these observations, it is proposed that iron can be taken up by the retinal EC via transferrin receptor-mediated endocytosis and exported to the retina via ferroportin. Retinal vasculature in the inner BRB protects the retina from fluctuations in systemic iron levels, and retinal EC plays crucial roles in the maintenance of local iron homeostasis in the retina. Systemic iron overload induced by intraperitoneal injection of iron dextran to adult mice resulted in iron trapping in retinal vasculature with minimal iron penetration into the retina [95]. 

Exported from retinal EC, iron is thought to be imported by glial cells expressing proteins involved in iron uptake, such as transferrin, transferrin receptor, and metal-ion transporters including Zip8 and Zip14 [107]. In the mouse retina, Müller cells have the highest expression of ceruloplasmin, a ferroxidase that converts Fe^2+^ to Fe^3+^ and facilitates iron export of ferroportin. Ceruloplasmin is critical for ferroportin to export iron. Homozygous ceruloplasmin knockout showed that iron remained in the ferroportin and ubiquitination-mediated ferroportin degradation [108]. The high expression of ceruloplasmin in Müller cells suggests that they play pivotal roles in retinal iron distribution by actively exporting iron with different types of retinal cells. Moreover, Müller cells are important in the maintenance of retinal iron homeostasis. Conditional ablation of Müller cells in mice resulted in blood–retinal barrier breakdown and iron accumulation throughout the neurosensory retina [107].

Exported by Müller cells, iron can be imported by photoreceptors expressing transferrin receptor, Zip8 and Zip14 [107]. In adult rat retinas, transferrin receptor 1 expression is more prominent in the photoreceptor inner segments than outer segments [100]. Moreover, the immunostaining of mouse retina showed the photoreceptor inner segments contain higher levels of DMT1, ferritin, and ferroportin than outer segments [67], which indicates that iron is more actively utilized in the inner segments. Iron in photoreceptors can be exported via ferroportin or phagocytosis of shed photoreceptor outer segments by RPE cells [109] (Figure 3C). Here, we propose the decreased hepcidin in the liver of *Cyp1b1*^−/−^ mice leads to increased systemic iron levels. We propose the increased systemic iron is sensed by retinal EC, resulting in increased intracellular iron in EC and perhaps in other ocular cell types with intact ferroportin–hepcidin axis, including trabecular meshwork cells, resulting in oxidative stress.

## 6. Iron and Oxidative Stress

Iron is crucial to normal cell metabolism, but Fe^2+^ can react with hydrogen peroxide (H_2_O_2_) generating hydroxyl radial (HO^•^), which is one of the most potent ROSs known via the Fenton reaction (Fe^2+^ + H_2_O_2_ → Fe^3+^ + ^•^OH + OH^−^, k = 76 M^−1^s^−1^) [110]. Fe^3+^ reacts with H_2_O_2_ producing perhydroxyl radical (HO_2_^•^) via Fenton-like reaction (Fe^3+^ + H_2_O_2_ → Fe^2+^ + HO_2_^•^ + H^+^, k = 0.01 M^−1^s^−1^), which is four magnitude orders slower than the Fenton reaction. The HO_2_^•^ is less potent than other ROS, but it reacts with Fe(III) and reduces it to Fe^2+^ (HO_2_^•^ + Fe^3+^ → O_2_ + H^+^ + Fe^2+^), which maintains the hydroxyl radical production by Fe^2+^ [111]. Fe^3+^ is also reduced to Fe^2+^ in the presence of superoxide and Fe^2+^ (Fe^3+^ + O_2_^•−^ → Fe^2+^ + O_2_). The net effect of the Fenton reaction and reduction in Fe^3+^ to Fe^2+^ is the generation of hydroxyl radical from hydrogen peroxide and superoxide catalyzed by iron ions (O_2_^•−^ + H_2_O_2_ → ^•^OH + OH^−^ + O_2_), which is called the Haber–Weiss reaction [112].

Hydroxyl radical (^•^OH) is the neutral form of the hydroxide ion (OH^−^) and, along with superoxide (O_2_^•−^), is one of the two major ROS in living organisms [113]. Due to its high 1-electron reduction potential, hydroxyl radical is one of the strongest chemical oxidizing agents [114]. Hydroxyl radicals react with almost every type of molecule in cells, including lipids, polypeptides, proteins, nucleic acids, and sugars [115]. Moreover, by extracting an electron from fatty acids, especially polyunsaturated fatty acids (PUFA), and forming lipid radical (L^•^), hydroxyl radicals mediate the initiation step of lipid peroxidation. The end products of lipid peroxidation of PUFA are electrophilic and reactive aldehydes, including malondialdehyde (MDA), 4-hydroxynonenal (4-HNE), propanal, and hexanal. These lipid peroxidation-derived aldehydes diffuse across membranes and covalently modify proteins and other macromolecules such as nucleic acids, lipids, and carbohydrates, resulting in loss of function of the molecules and/or forming adducts. Among the lipid production products, MDA is the most capable of damaging the DNA, and 4-HNE is the most toxic product [116]. 

Increased oxidative stress generally occurs when the cellular antioxidant system is overwhelmed with excess levels of ROS. This leads to cellular damage and death. The enzymatic and non-enzymatic antioxidant systems are integrated to stabilize or inactivate ROS before interacting with cellular components. The combination of antioxidant and membrane repair systems provides cells with a defense against various insults and cell death. Glutathione, a cysteine-containing tripeptide, is recognized as the key intracellular antioxidant. Its level is controlled by the availability of intracellular cysteine, which is mainly provided through the System Xc^−^. An amino acid transporter, System Xc^−^, consists of two protein subunits, namely SLC7A11 and SLC3A2, and imports cystine and exports glutamate. Cysteine, the limiting precursor for GSH production, is generated by reduction in the intracellular cysteine. Cysteine may also be produced through the activation of the trans-sulfation pathway, the major route for the metabolism of sulfur-containing amino acids. 

Lipid peroxidation can damage various cellular components, including cellular membranes, peptides, and nucleic acids. Reactive oxygen species induced by lipid peroxidation is associated with different forms of cell death, including apoptosis, autophagy, and ferroptosis [117]. Ferroptosis is a form of cell death mediated by increased iron levels and enhanced lipid peroxidation driving the rupture of plasma membrane and cell death. GSH is necessary for the activity of the GPX4, the major repressor of ferroptosis, reducing toxic phospholipid hydroperoxides to non-toxic phospholipid alcohols. Our knowledge regarding the role and regulation of ferroptosis mainly comes from the use of System Xc^−^ inhibitors, including erastin, sulfasalazine, and sorafenib, or GPX4 inhibitors, including RSL3, ML162, and ML210. N-acetylcysteine is a precursor for glutathione production and GPX4 activity, and its ability to effectively negate the adverse effects of CYP1B1 deficiency on retinal vasculature and trabecular meshwork is consistent with the proposed role of excess iron accumulation and oxidative stress, as well as the mitigation of retinal neovascularization and dysgenesis of the trabecular meshwork.

## 7. CYP1B1 and Regulation of Iron Homeostasis

CYP1B1 expression in the eye is an important modulator of developmental processes. Mutations in CYP1B1 are associated with the development of primary congenital glaucoma in humans [118]. However, the underlying molecular and cellular mechanisms that regulate CYP1B1 expression and activity in these processes remain unknown. Mice deficient in CYP1B1 exhibit defects in the development and function of the conventional outflow pathway, including the trabecular meshwork and Schlemm’s canal. This defect was exacerbated in albino mice, suggesting a role for increased oxidative stress in these processes [119]. The ability of melanosomes to bind iron within living cells could contribute to the protection noted in pigmented mice [120]. Although we have noted significant constitutive expression of CYP1B1 in various ocular cell types, its expression is also induced by exposure to AhR agonists such as dioxin [13]. However, the normal regulatory mechanisms that keep CYP1B1 expression in check remain poorly understood. There have been numerous efforts toward identifying physiological substrates and metabolites of CYP1B1 in order to advance our knowledge regarding the molecular and cellular mechanisms that impact CYP1B1 expression, activity, and function. We have shown that CYP1B1 expression has a significant impact on adhesive and migratory properties of various ocular cell types, including retinal vascular and trabecular meshwork cells [121]. How CYP1B1 expression and/or activity modulate these cell properties, which are likely linked to cellular redox state, needs further exploration.

CYPs are generally considered as liver resident enzymes whose activation by exposure to various toxicants, including aromatic hydrocarbons, drive the detoxification and elimination of these chemicals. With advancements in genomic and transcriptomic studies, it is now recognized that CYP expression and activity play significant roles in the modulation of proper developmental and metabolic functional activities of various cell types and tissues. Estradiol and arachidonic acid are key endogenous substrates of CYP1B1, whose metabolites have significant impacts on various biological functions. CYP1B1 expression also has important roles in retinoic acid metabolism during early developmental processes. CYP1B1 is also involved in the metabolism of exogenous toxic chemicals, enhancing their elimination and minimizing their adverse effects on human health (reviewed in [21]).

Estrogens are metabolically converted to estrogenically inactive metabolites for elimination from the body. The hydroxylation of estrogens by CYP enzymes is the first step in their metabolism, mainly in the liver. 2-hydroxyestradiol, mainly catalyzed by CYP1A2 and CYP3A4 in the liver, and CYP1A1 in extrahepatic tissues, is the major metabolite of estradiol. However, in the estrogen target tissues which express a high level of CYP1B1; 4-hydroxyestradiol is the predominant estradiol metabolite [122]. 4-hydroxyestradiol produces free radicals from the oxidative-reductive cycling with the corresponding semiquinone and quinone forms, which cause cellular damage and could have adverse impact on tissue integrity and function [122]. In addition, estradiol regulates human CYP1B1 expression through estrogen receptor alpha [123]. Thus, the regulation of estrogen-metabolizing CYP enzymes by estrogen itself could contribute to local homeostasis of estrogens. Thus, the absence of CYP1B1 expression or inhibition of its activity may lead to accumulation of estradiol and enhanced estrogen receptor signaling, the consequences of which remain largely unexplored. A recent study by Kurmann et al. showed that estradiol inhibits the migratory activity of brain vascular pericytes [124]. Estradiol enhanced the barrier function of endothelial cells when cocultured with pericytes, which likely accounts for estradiol protection against blood–brain barrier disruption and antiangiogenic activity [124]. This notion is supported by our studies demonstrating the mitigation of angiogenesis in *Cyp1b1*^−/−^ mice [34], likely because of increased levels of estradiol in these mice. How these changes in estradiol metabolism impact the cellular redox state and increased oxidative stress that we noted in various tissues of *Cyp1b1*^−/−^ mice needs further exploration.

The protective impact of estradiol on cardiovascular integrity and function has been known for quite some time. However, the underlying mechanisms and the cell-autonomous impact of estradiol on vascular cells are beginning to provide novel insight into the mechanisms involved, as discussed above. Estradiol signals through estrogen receptor in EC and regulates their proangiogenic properties [125,126]. One of the genes whose expression is increased by estradiol is BMP6 [127,128]. However, little is known about the autocrine and/or paracrine signaling of BMP6 in the retinal vasculature. BMP6 produced by liver SEC drives the expression of hepcidin in hepatocytes regulating systemic iron levels [129]. Given that retinal EC are the major regulators of iron homeostasis in the retina and express key iron regulatory proteins, we propose that enhanced accumulation of estradiol in the absence of CYP1B1 could lead to increased BMP6 production by retinal EC and altered expression of hepcidin altering local iron levels. We hypothesize that increased hepcidin expression in retinal EC diminishes ferroportin expression in these cells, thus increasing intercellular iron levels leading to increased oxidative stress, lipid peroxidation, and cell death. This would mitigate angiogenesis and drive trabecular meshwork dysgenesis in *Cyp1b1*^−/−^ mice as we have previously demonstrated [32,34]. These possibilities are being presently explored in our laboratory and will provide novel insight into the regulatory pathways which mediate CYP1B1 activity in the eye (Figure 5). The impact of hepcidin on ferroportin expression in bovine retinal EC culture demonstrated decreased ferroportin expression and diminished iron export [130].

CYP1B1 is also an important regulator of fatty acid homeostasis, and its expression is important in the modulation of PPARɣ and PPARα target genes and fatty acid metabolism. Arachidonic acid (AA) is oxidized by human CYP1B1 generating hydroxy eicosatetraenoic acids (HETEs), including 20-HETE and 12-HETE. However, the metabolism of AA by mouse CYP1B1 generates epoxy eicosatetraenoic acids (EETs) [21]. These metabolites impact various cellular and tissue functions, including cardiovascular, growth, apoptosis, vasodilation, and fibrosis. The differences in the catalytic efficiency of human and mouse CYP1B1 might contribute to their differences in AA metabolism. The potential pathophysiological impact of these AA metabolites will benefit from further exploration of their functions. We are conducting untargeted metabolomics studies to identify additional putative endogenous substrates of CYP1B1 with important physiological functions.

Vitamin A is critical for growth and development and supports the health of the immune system and vision. Retinol, retinal, and retinoic acid are the major forms of vitamin A. CYP1B1 catalyzes the oxidative metabolism of retinol to retinal and retinal to retinoic acid. Although the oxidation of retinol to retinoic acid is regulated by both human and mouse CYP1B1, neither can oxidize retinoic acid [21]. The interaction of retinoic acid with its receptors, including retinoic acid receptors and retinoid-X receptor (RXR), initiates signaling pathways with positive impacts on dyslipidemia, atherosclerosis, and cancer [131,132]. We previously showed retinol, all-trans retinoic acid, and AM580 (an RXR agonist) failed to overcome the impact of CYP1B1 deficiency on retinal EC proangiogenic activity [34]. However, further delineating the impact of vitamin A metabolites on vision would be informative. 

## 8. Conclusions

CYP1B1 mediated NADPH-dependent phase I metabolism of a variety of xenobiotics. CYP1B1 can activate a series of environmental carcinogens through hydroxylation of their procarcinogens. The role of CYP1B1 in carcinogen activation is supported by reduced carcinogenesis noted in *Cyp1b1*^−/−^ mice. Thus, CYP1B1 is an important activator of environmental carcinogens. Xenobiotics, such as TCDD, also induce the expression of CYP1B1 through AhR, further enhancing their oxidation. CYP1B1 is highly expressed in many types of cancers, and its mutated alleles are detected in cancer and glaucoma. Thus, regulation of CYP1B1 expression and/or activity could be a suitable target for treatment strategies. Thus far, many natural and synthetic compounds that inhibit CYP1B1 activity have been identified. These compounds generally fall into four categories, including stilbene, flavonoids, coumarin, and anthraquinone. Some of these inhibitors also antagonize AhR expression and suppress CYP1B1 expression, making the interpretation of outcomes of CYP1B1 inhibition by these compounds challenging. The use of CYB1B1 inhibitors will allow further delineation of CYP1B1 function and potentially target it for therapeutic intervention.

We showed the inhibition of CYP1B1 activity with tetra-methoxy stilbene (TMS) in wild-type cells causes increased oxidative stress and mitigates proangiogenic activity of vascular cells, as noted in cells prepared from *Cyp1b1*^−/−^ mice. Furthermore, we showed the administration of N-acetylcysteine to *Cyp1b1*^−/−^ mice protected trabecular meshwork dysgenesis and restored retinal neovascularization during oxygen-induced ischemic retinopathy. These observations established an important role for CYP1B1 in the modulation of cellular redox state, such that its absence or inhibition results in increased oxidative stress. Our recent studies are focused on delineating how a lack of CYP1B1 expression and/or activity could result in oxidative stress. Thus far, our findings suggest an important role for CYP1B1 in the regulation of systemic and local iron homeostasis through changes in BMP6 and hepcidin expression. However, the underlying mechanisms involved remain unresolved and need further exploration.

We propose a lack of CYP1B1 may result in accumulation of its endogenous substrates such as estradiol and AA. The significant role of estradiol metabolites generated by CYP1B1 and their cardioprotective action has been recognized [133]. The action of estradiol through its receptor (ERα present on EC) could affect the BMP6–hepcidin–ferroportin axis in retinal EC. Estradiol treatment of hepG2 cells resulted in increased expression of BMP6 and hepcidin production [127]. The autocrine action of hepcidin could result in the degradation of ferroportin and enhanced iron accumulation and oxidative stress in retinal EC, as we noted in *Cyp1b1*^−/−^ retail EC [34]. We know this increased oxidative stress can be relieved by administration of N-acetylcysteine and likely by iron chelators. Given the fact that N-acetylcysteine and/or iron chelators are currently in clinical use, their utilization to prevent the development and progression of primary congenital glaucoma, as we showed in *Cyp1b1*^−/−^ mice, is well worth exploring.

Our gene expression studies using livers from *Cyp1b1*^−/−^ mice demonstrated important roles for CYP1B1 expression in the modulation of various metabolic activity and systemic iron homeostasis, likely through the accumulation of its endogenous substrates such as estradiol and AA. We noted that a lack of Cyp1b1 expression was associated with a decreased level of hepcidin expression, most likely through the downregulation of BMP6 production and activity of liver SEC (Figure 1). This is expected to result in increased systemic iron levels and likely altered tissue-specific iron homeostasis. Increased systemic iron levels could induce hepcidin production locally, which decreases ferroportin level. This could result in enhanced intracellular accumulation of iron in cells that express ferroportin, such as retinal EC and trabecular meshwork cells. Although studies on the liver suggest a paracrine mechanism for the expression of hepcidin produced by EC and downregulation of ferroportin in the cells with iron-storing capacity, such as intestinal cells, we believe this could also occur locally in an autocrine fashion, as shown to occur in retinal EC downregulating ferroportin levels and enhancing iron accumulation [95,96]. Although this is initially used to protect the neuroretina from excess systemic iron, it is unclear how this is maintained with chronic increases in systemic iron levels. It is this chronic systemic and local regulation of iron homeostasis that ultimately leads to tissue damage and dysfunction. More studies are needed to further investigate how CYP1B1 activity regulates systemic and local tissue-specific iron homeostasis and oxidative stress to maintain tissue integrity and function.

## Figures and Tables

**Figure 1 cells-11-02930-f001:**
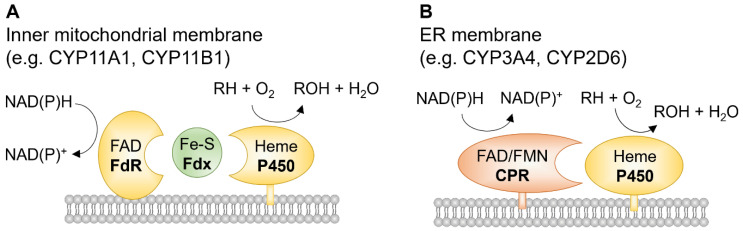
Two major mammalian cytochrome P450 systems with redox partner proteins. (**A**) The mitochondrial P450 systems belong to the class I system, consisting of three proteins: flavin adenine dinucleotide (FAD)-containing ferredoxin reductase (FdR), ferredoxin (Fdx/Fe-S), and P450. (**B**) class II P450 systems are most common in mammalian cells. NADPH-cytochrome P450 reductase (CPR) contains FAD and flavin mononucleotide (FMN) as prosthetic groups. Adapted from [8].

**Figure 2 cells-11-02930-f002:**
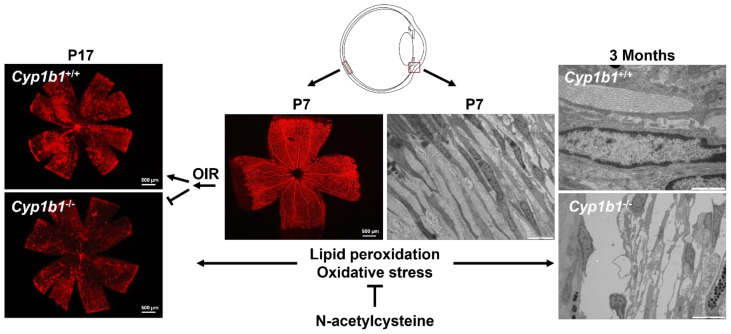
Cyp1b1 expression and regulation of oxidative stress in ocular tissues. At P7, *Cyp1b1*^−/−^ mice showed normal retinal vasculature and trabecular meshwork (TM). During oxygen-induced ischemic retinopathy (OIR), *Cyp1b1*^−/−^ mice showed attenuated neovascularization at P17 with increased oxidative stress (Left). *Cyp1b1*^−/−^ mice at 3 months of age showed degeneration of trabecular collagen fibrils and TM dysgenesis, which progressively worsens with age and increased oxidative stress (Right). Administration of antioxidant N-acetylcysteine protected TM from degeneration and restored neovascularization in *Cyp1b1*^−/−^ mice [32,34].

**Figure 4 cells-11-02930-f004:**
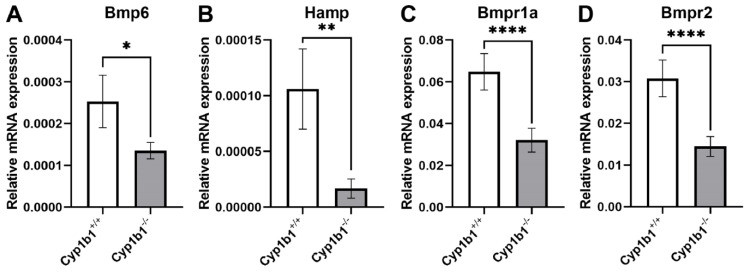
Decreased BMP6 and hepcidin gene expression levels in *Cyp1b1*^−/−^ liver sinusoidal endothelial cells (SEC). RNA was prepared from liver SEC isolated from wild-type (*Cyp1b1*^+/+^) and Cyp1b1-deficient (*Cyp1b1*^−/−^) mice as previously described by us [43]. qPCR analysis was performed to compare Bmp6 (**A**), Hamp (**B**), Bmpr1a (**C**), and Bmpr2 (**D**) in *Cyp1b1*^+/+^ and *Cyp1b1*^−/−^ liver SEC. Samples were conducted in triplicate and repeated twice using different isolations of liver SEC (**p* < 0.05, ***p* < 0.01, *****p* < 0.0001).

**Figure 5 cells-11-02930-f005:**
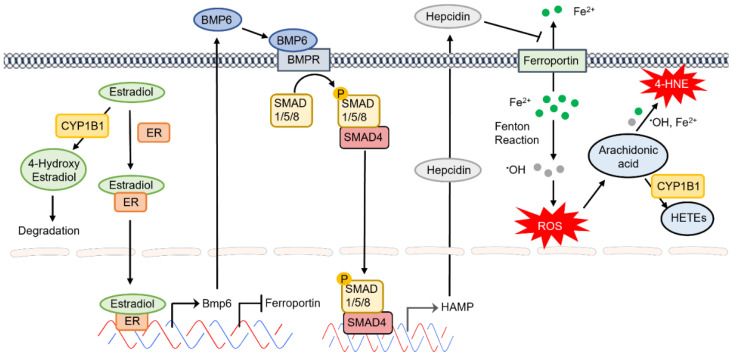
The proposed CYP1B1 regulation of intracellular iron levels and oxidative stress through estradiol metabolism, BMP6 signaling, hepcidin production, and ferroportin inhibition in the retinal endothelium.

**Table 1 cells-11-02930-t001:** Cytochrome P450 families and their main functions. Adapted from [7].

Family	Members	Subfamily (genes)	Main function
CYP1	3 subfamilies, 3 genes1 pseudogene	A (CYP1A1, CYP1A2), B (CYP1B1)	Xenobiotic and steroid (including estrogen) metabolism
CYP2	13 subfamilies, 16 genes 16 pseudogenes	A (CYP2A6, CYP2A7, CYP2A13), B (CYP2B6), C (CYP2C8, CYP2C9, CYP2C18, CYP2C19), D (CYP2D6), E (CYP2E1), F (CYP2F1), J (CYP2J2), R (CYP2R1), S (CYP2S1), U (CYP2U1), W (CYP2W1)	Xenobiotic and steroid metabolism
CYP3	1 subfamiliy, 4 genes 2 pseudogenes	A (CYP3A4, CYP3A5, CYP3A7, CYP3A43)	Xenobiotic and steroid (including testosterone) metabolism
CYP4	6 subfamilies, 12 genes 10 pseudogenes	A (CYP4A11, CYP4A22), B (CYP4B1), F (CYP4F2, CYP4F3, CYP4F8, CYP4F11, CYP4F12, CYP4F22), V (CYP4V2), X (CYP4X1), Z (CYP4Z1)	Fatty acid, arachidonic acid, and leukotriene metabolism
CYP5	1 subfamiliy, 1 gene	A (CYP5A1)	Thromboxane A2 synthesis
CYP7	2 subfamilies, 2 genes	A (CYP7A1), B (CYP7B1)	Bile acid synthesis
CYP8	2 subfamilies, 2 genes	A (CYP8A1), B (CYP8B1)	Prostacyclin syntehsis, bile acid syntehsis
CYP11	2 subfamilies, 3 genes	A (CYP11A1), B (CYP11B1, CYP11B2)	Steroid synthesis
CYP17	1 subfamiliy, 1 gene	A (CYP17A1), B (CYP17B1)	Steroid synthesis
CYP19	1 subfamiliy, 1 gene	A (CYP19A1)	Steroid synthesis
CYP20	1 subfamiliy, 1 gene	A (CYP20A1)	Unknown
CYP21	2 subfamilies, 1 gene 1 pseudogene	A (CYP21A2)	Steroid synthesis
CYP24	1 subfamiliy, 1 gene	A (CYP24A1)	Vitamin D metabolism
CYP26	3 subfamilies, 3 genes	A (CYP26A1), B (CYP26B1), C (CYP26C1)	Vitamin A metabolism
CYP27	3 subfamilies, 3 genes	A (CYP27A1), B (CYP27B1), C (CYP27C1)	Vitamin D and bile acid synthesis
CYP39	1 subfamiliy, 1 gene	A (CYP39A1)	Bile acid synthesis
CYP46	1 subfamiliy, 1 gene	A (CYP46A1)	Cholesterol synthesis
CYP51	1 subfamiliy, 1 gene	A (CYP51A1)	Cholesterol synthesis

## Data Availability

Not applicable.

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
