# Peer review of "Cytochrome P450 1B1: A Key Regulator of Ocular Iron Homeostasis and Oxidative Stress"

_cells, 2022, doi:10.3390/cells11192930_

Round 1
Reviewer 1 Report
Comments:
In the present manuscript, Song et al., reviewed many aspects that connect the Cytochrome P450 1B1 functions and activities and iron metabolism, with particular focus on the retina.
The work sounds interesting, with a comprehensive and complete overview of the present knowledge of the field.
The review is properly written, and the references introduced fit with the text.
However, the reviewer has some suggestions.
Major suggestions
It would be helpful, in this quite long and detailed review, to introduce some more figures.
For example, it could be helpful for the reader:
- a summary table with the main classification of CYPs (Paragraph 1);
- a diagram or graph with the main functions of CYPs;
- a representative figure for the local iron homeostasis in the retina, with the different cells and tissue involved (Paragraph 6);
- a representative figure related to the CYP1B1 in the ocular tissue and its connection with the iron homeostasis (paragraph 8);
- …
In the second paragraph (2. CYP1B1), the part related to iron homeostasis and CYP1B1 (from line 118 till the end of the paragraph) could be moved in the Paragraph 8, which is properly dedicated to CYP1B1 and iron. It could be easier for the reader first to have a description of CYPs and CYP1B1 in general, then to have an overview of iron homeostasis and finally read about the connection between the two fields. It could be more linear and less confusing.
The last Paragraph (9. Conclusion) could be better written, trying to summarize the previous paragraphs and pointing out the questions remained still unsolved, which in part they are present along all the text, but they could be summarized in the last Paragraph.
Minor suggestions
Some minor mistakes:
- The “1. Introduction” is written in a smaller character that the other titles and not in bold;
- Define the acronymous with the extended name, as first mentioned (example: EC);
- Line 396: protein is mentioned twice. Delete one;
- Line 539: correct CY1B1 with CYP1B1.
Author Response
In the present manuscript, Song et al., reviewed many aspects that connect the Cytochrome P450 1B1 functions and activities and iron metabolism, with particular focus on the retina.
The work sounds interesting, with a comprehensive and complete overview of the present knowledge of the field. The review is properly written, and the references introduced fit with the text. However, the reviewer has some suggestions. Thank you for the positive comments.
Major suggestions
It would be helpful, in this quite long and detailed review, to introduce some more figures. For example, it could be helpful for the reader:
- a summary table with the main classification of CYPs (Paragraph 1): Please see table 1.
- a diagram or graph with the main functions of CYPs: Please see Figure 1.
- a representative figure for the local iron homeostasis in the retina, with the different cells and tissue involved (Paragraph 6); Please see Figure 3
- a representative figure related to the CYP1B1 in the ocular tissue and its connection with the iron homeostasis (paragraph 8): please see figure 2.
- …
In the second paragraph (2. CYP1B1), the part related to iron homeostasis and CYP1B1 (from line 118 till the end of the paragraph) could be moved in the Paragraph 8, which is properly dedicated to CYP1B1 and iron. It could be easier for the reader first to have a description of CYPs and CYP1B1 in general, then to have an overview of iron homeostasis and finally read about the connection between the two fields. It could be more linear and less confusing. We now have revised the text as suggested.
The last Paragraph (9. Conclusion) could be better written, trying to summarize the previous paragraphs and pointing out the questions remained still unsolved, which in part they are present along all the text, but they could be summarized in the last Paragraph. The discussion has been revised and expanded to address the reviewer comments.
Minor suggestions
Some minor mistakes:
- The “1. Introduction” is written in a smaller character that the other titles and not in bold; Corrected.
- Define the acronymous with the extended name, as first mentioned (example: EC); Done.
- Line 396: protein is mentioned twice. Delete one; Done.
- Line 539: correct CY1B1 with CYP1B1. Done.
Reviewer 2 Report
Yong-Seok Song1, et al. present an interesting study that evaluates Cytochrome P450 1B1: A key Regulator of Ocular Iron Homeostasis and Oxidative Stress
The study results certainly suggest some degrees that the important role CYP1B1 expression and 39 activity plays in maintaining retinal redox homeostasis through modulation of iron levels by retinal endothelial cells. The relationship between CYP1B1 expression and activity and 41
iron levels have not been previously delineated
Besides how the magnitude of these data add new findings compare to the current standard can not be determined based on this study
The results are encouraging and further study is warranted.
here some relevant points :
1. please add on the keywords this does not match with the manuscript
2. The authors should express why is relevant for RD and DME patients evaluatesCytochrome P450 1B1: A key Regulator of Ocular Iron Homeostasis and Oxidative Stress
What does it Change for the current standard of care ?
3. The authors should explain why their findings make a different for ophthalmologist around the world and for the readers of Cells
4. The authors should explain the source of the information and what were the criteria they used for adding to the paper Were the assessors masked? What was the ICC between them in orden to analyzed the data ? I Was the randomization digitalized? . If not please added
5. Please add references and rephrase the sentence. English grammar should be applied.
“The retina is normally protected from changes in the systemic circulation through 150 the retinal-blood barrier established by the retinal vascular EC. The retinal vascular EC 151 express molecular mediators of iron regulation such as hepcidin and ferroportin and are 152 key determinants of local iron homeostasis in the retina [28]. The increase in systemic iron 153 decreases ferroportin in retinal EC perhaps through up-regulation BMP6 expression by 154 retinal EC [29]. Furthermore, the intraocular administration of BMP6 protein increases ret- 155 inal hepcidin and alters retinal labile iron levels [30]. Iron is an essential element for various 156 cellular functions due to its ability to transfer electrons, but it catalyzes the formation of 157 hydroxyl radicals from hydrogen peroxide via the Fenton reaction (Fe2+ + H2O2 158 → Fe3+ + ·OH + OH-). In excess, iron can lead to accumulation of lipid peroxidation gen- 159 erating reactive oxygen species (ROS) and oxidative stress. Thus, ocular iron levels are 160 tightly regulated to maximize iron supply and avoid ROS generation. Alterations in iron 161 homeostasis, such as iron accumulation and iron deficiency, which impact oxidative stress 162 can lead to cell and tissue damage and dysfunction. “
‘
6. please add in the introduction papers which have been published showing the important for DME and RD patients to performed the test ancillary test such us OCTs , wield Field Retinography and add how this will help physician around the world to proper diagnosis not only for DME but also for macular deseases , add one line in the introduction of this and also in the discussion section These papers should be describe in this general considerations.
Reference these :
-Acta Diabetol. 2018 Jun;55(6):541-547. doi: 10.1007/s00592-018-1117-z.
Epub 2018 Mar 1.
Progression of diabetic retinopathy severity after treatment with
dexamethasone implant: a 24-month cohort study the 'DR-Pro-DEX Study’
--Retina. 2019 Jan;39(1):44-51. doi: 10.1097/IAE.0000000000002196.
Naïve subretinal haemorrhage due to neovascular age-related macular degeneration. pneumatic displacement, subretinal air, and tissue plasminogen activator: subretinal vs intravitreal aflibercept-the native study
Please apply correction for misspelling and English grammar . This paper should be corrected by an english redactor
Author Response
Comments and Suggestions for Authors
Yong-Seok Song1, et al. present an interesting study that evaluates Cytochrome P450 1B1: A key Regulator of Ocular Iron Homeostasis and Oxidative Stress
The study results certainly suggest some degrees that the important role CYP1B1 expression and activity plays in maintaining retinal redox homeostasis through modulation of iron levels by retinal endothelial cells. The relationship between CYP1B1 expression and activity and iron levels have not been previously delineated.
Besides how the magnitude of these data add new findings compare to the current standard cannot be determined based on this study. The results are encouraging and further study is warranted. We agree. This is an overview/review of what has been reported in the literature and what we have found so far as well as where we think the field should go.
here some relevant points:
- please add on the keywords this does not match with the manuscript. We have added some more relevant additional key words.
- The authors should express why is relevant for RD and DME patients evaluatesCytochrome P450 1B1: A key Regulator of Ocular Iron Homeostasis and Oxidative Stress
What does it Change for the current standard of care? There is no information available regarding the role CYP1B1 may play in RD and DME, and the phenotypes manifested by CYP1B1 deficiency is generally associated with higher levels of oxygen (>10%). That is why CYP1B1 does not affect any aspects of embryonic development in mice because the majority of developmental processes occur in relatively ischemic/low oxygen levels.
- The authors should explain why their findings make a different for ophthalmologist around the world and for the readers of Cells. CYP1B1 is one of the first CYPs identified to have important function in developmental processes, and its mutations in human accounts for the highest percentage of primary congenital glaucoma. Our work has clearly established a role for CYP1B1 in modulation of cellular redox state such that all cells we have looked at that lack CYP1B1 have higher oxidative stress. However, the impacts are tissue/cell specific.
- The authors should explain the source of the information and what were the criteria they used for adding to the paper Were the assessors masked? What was the ICC between them in orden to analyzed the data ? I Was the randomization digitalized? . If not please added. This is a review based on our studies and findings and proposing the potential underlying mechanisms that accounts for the changes we noted with CYP1B1 deficiency based on what is known in the literature.
- Please add references and rephrase the sentence. English grammar should be applied.Done.
“The retina is normally protected from changes in the systemic circulation through 150 the retinal-blood barrier established by the retinal vascular EC. The retinal vascular EC 151 express molecular mediators of iron regulation such as hepcidin and ferroportin and are 152 key determinants of local iron homeostasis in the retina [28]. The increase in systemic iron 153 decreases ferroportin in retinal EC perhaps through up-regulation BMP6 expression by 154 retinal EC [29]. Furthermore, the intraocular administration of BMP6 protein increases ret- 155 inal hepcidin and alters retinal labile iron levels [30]. Iron is an essential element for various 156 cellular functions due to its ability to transfer electrons, but it catalyzes the formation of 157 hydroxyl radicals from hydrogen peroxide via the Fenton reaction (Fe2+ + H2O2 158 → Fe3+ + ·OH + OH-). In excess, iron can lead to accumulation of lipid peroxidation gen- 159 erating reactive oxygen species (ROS) and oxidative stress. Thus, ocular iron levels are 160 tightly regulated to maximize iron supply and avoid ROS generation. Alterations in iron 161 homeostasis, such as iron accumulation and iron deficiency, which impact oxidative stress 162 can lead to cell and tissue damage and dysfunction. This is done as suggested.
- please add in the introduction papers which have been published showing the important for DME and RD patients to performed the test ancillary test such us OCTs , wield Field Retinography and add how this will help physician around the world to proper diagnosis not only for DME but also for macular deseases , add one line in the introduction of this and also in the discussion section These papers should be describe in this general considerations.
Reference these :
-Acta Diabetol. 2018 Jun;55(6):541-547. doi: 10.1007/s00592-018-1117-z.
Epub 2018 Mar 1.
Progression of diabetic retinopathy severity after treatment with
dexamethasone implant: a 24-month cohort study the 'DR-Pro-DEX Study’
--Retina. 2019 Jan;39(1):44-51. doi: 10.1097/IAE.0000000000002196.
DEXAMETHASONE IMPLANT FOR DIABETIC MACULAR EDEMA IN NAIVE COMPARED WITH REFRACTORY EYES: The International Retina Group Real-Life 24-Month Multicenter Study. The IRGREL-DEX Study. -Eye .2022 Aug 29. doi: 10.1038/s41433-022-02222-z. Online ahead of print.
Naïve subretinal haemorrhage due to neovascular age-related macular degeneration. pneumatic displacement, subretinal air, and tissue plasminogen activator: subretinal vs intravitreal aflibercept-the native study.
We apologize for not clearly understanding what the reviewer’s concerns are since we do not discuss anything related to the utility of OCT and clinical evaluations in relation to DME and RD. This is a mechanistic review regarding how Cyp1b1 expression and/or activity could modulate cellular oxidative stress and what role changes in iron homeostasis may play in the noted oxidative stress in response to CYP1B1 deficiency.
Please apply correction for misspelling and English grammar. This paper should be corrected by an english redactor. Done
Reviewer 3 Report
The review by Song et al focuses on the potential role of CYP1B1 in the regulation of ocular iron homeostasis and oxidative stress. The manuscript summarizes available data about the expression and function of CYP1B1, concentrating on data obtained from previous work in Cyp1b1-/- mice. Then it provides a detailed description of systemic and local (liver and retina) iron homeostasis; finally it presents the possible relationship between CYP1B1 and iron metabolism in the retina, speculating on the molecular mechanisms linking the monooxygenase with the control of oxidative stress and angiogenic properties in retinal Endothelial cells.
Although the topic is of interest and its development may lead to a better understanding of the mechanisms controlling the physiopathology of iron metabolism, especially in the retina, this reviewer has major concerns about the timing and robustness of the review:
1) As stated in the title, the review suggests that CYP1B1 is a controller of iron homeostasis and oxidative stress in the eye. The data supporting this concept are clearly not conclusive and very little is known about the underlining mechanisms. This is clear by reading the section discussing CYP1B1 and the regulation of iron homeostasis, where the authors repeat for several times that the issue needs further exploration. At the moment, the connection between CYP1B1 and iron homeostasis is mainly supported by the changes in BMP6 and Hepcidin mRNA level observed in CYP1B1 KO mice. I think that more robust demonstration is needed before starting to speculate about possible biological mechanisms involved in the process.
2) There are several, extensive and update reviews in the literature regarding systemic and cellular iron homeostasis. The sections about this topic (a large part of the review) are not adding new information and many details are not required to understand and support the potential connection between iron and CYP1B1
3) I do not understand why the authors are presenting the data in figure 1. The changes in Bmp6 and hepcidin expression are already published. Those about Bmp receptors are new, but do not add much to the initial hypothesis and should be presented in an experimental work further investigating this issue.
In conclusion, I believe that the line of research discussed in the manuscript is of high interest, but available data are too limited and premature to support the conclusion presented in the title and in chapter 9 and the several speculative mechanistic hypothesis raised by the authors.
Author Response
The review by Song et al focuses on the potential role of CYP1B1 in the regulation of ocular iron homeostasis and oxidative stress. The manuscript summarizes available data about the expression and function of CYP1B1, concentrating on data obtained from previous work in Cyp1b1-/- mice. Then it provides a detailed description of systemic and local (liver and retina) iron homeostasis; finally it presents the possible relationship between CYP1B1 and iron metabolism in the retina, speculating on the molecular mechanisms linking the monooxygenase with the control of oxidative stress and angiogenic properties in retinal Endothelial cells.
Although the topic is of interest and its development may lead to a better understanding of the mechanisms controlling the physiopathology of iron metabolism, especially in the retina, this reviewer has major concerns about the timing and robustness of the review:
1) As stated in the title, the review suggests that CYP1B1 is a controller of iron homeostasis and oxidative stress in the eye. The data supporting this concept are clearly not conclusive and very little is known about the underlining mechanisms. This is clear by reading the section discussing CYP1B1 and the regulation of iron homeostasis, where the authors repeat for several times that the issue needs further exploration. At the moment, the connection between CYP1B1 and iron homeostasis is mainly supported by the changes in BMP6 and Hepcidin mRNA level observed in CYP1B1 KO mice. I think that more robust demonstration is needed before starting to speculate about possible biological mechanisms involved in the process. We do agree with what the reviewer indicating. However, our data and publications regarding the impact of Cyp1b1 deficiency on cellular oxidative stress are solid. We have both in vitro (cell culture) and in vivo (preclinical models). Our studies in the liver suggest decreased hepcidin expression could have a significant impact on systemic, and likely local, iron homeostasis. The role of iron in oxidative stress and tissue damage has been extensively studied and reported. We are proposing that CYP1B1 could mediate a significant part of its activity through modulation of iron homeostasis through its metabolic activity. We agree additional studies needed and this is what we are laying down here to exactly address the points raised by the reviewer. Since this is a review article we feel some speculation and suggestion of proposed studies to address questions raised are reasonable.
2) There are several, extensive and update reviews in the literature regarding systemic and cellular iron homeostasis. The sections about this topic (a large part of the review) are not adding new information and many details are not required to understand and support the potential connection between iron and CYP1B1. We agree and cite where key pathways have been previously reviewed. However, we felt in order to put our findings into perspective it will be useful to walk the reader through what is known and how our story fit with what is known. Prior to this review there has been no discussion of CYP1B1 relationship with regulation of iron homeostasis that we are aware of.
3) I do not understand why the authors are presenting the data in figure 1. The changes in Bmp6 and hepcidin expression are already published. Those about Bmp receptors are new, but do not add much to the initial hypothesis and should be presented in an experimental work further investigating this issue. We have fixed the text to better address our point. The gene array studies suggested down regulation of hepcidin in the liver from CYP1b1-/- mice but these data were not confirmed and also whether EC were responsible for these changes. Since we had isolated the liver EC from WT and Cyp1b1-/- mice we provide additional rigor for array studies.
In conclusion, I believe that the line of research discussed in the manuscript is of high interest, but available data are too limited and premature to support the conclusion presented in the title and in chapter 9 and the several speculative mechanistic hypothesis raised by the authors. This was not our intention and we point out on several occasions that our hypothesis requires future evaluation. We have tried to make this clearer for the reader.
Round 2
Reviewer 3 Report
The authors replied to the criticisms raised by the other reviewers. Even though the style and organization of the manuscript have been improved, the issues I raised in my first review still persist.